# Modular assembly of indole alkaloids enabled by multicomponent reaction

Jiaming Li[1], Zhencheng Lai[1], Weiwei Zhang[1], Linwei Zeng [1] & Sunliang Cui [1] ✉

Indole alkaloids are one of the largest alkaloid classes, proving valuable structural moiety in pharmaceuticals. Although methods for the synthesis of indole alkaloids are constantly explored, the direct single-step synthesis of these chemical entities with broad structural diversity remains a formidable challenge. Herein, we report a modular assembly of tetrahydrocarboline type of indole alkaloids from simple building blocks in a single step while showing broad compatibility with medicinally relevant functionality. In this protocol, the 2-alkylated or 3-alkylated indoles, formaldehyde, and amine hydrochlorides could undergo a one-pot reaction to deliver γ-tetrahydrocarbolines or β-tetrahydrocarbolines directly. A wide scope of these readily available starting materials is applicable in this process, and numerous structural divergent tetrahydrocarbolines could be achieved rapidly. The control reaction and deuterium-labelling reaction are conducted to probe the mechanism. And mechanistically, this multicomponent reaction relies on a multiple alkylamination cascade wherein an unusual $C(sp^3)-C(sp^3)$ connection was involved in this process. This method could render rapid access to pharmaceutically interesting compounds, greatly enlarge the indole alkaloid library and accelerate the lead compound optimization thus facilitating drug discovery.

Indole alkaloids, one of the largest classes of naturally occurring alkaloids, have proven to be an important and rich source of pharmaceuticals[1–8]. Particularly, tetrahydrocarbolines are the common structural motif in most of the indole alkaloids and represent a key scaffold in a wide range of market drugs and clinical candidates[9–13], such as hypotensive drug reserpine, antiphosphodiesterase-5 agent Tadalafil, and antiallergic therapeutics mebhydrolin and dimebolin (Fig. 1a)[14–18]. Therefore, the study of tetrahydrocarboline synthesis and drug discovery has inspired practitioners from a wide spectrum of the scientific community. Typically, the β-tetrahydrocarbolines biogenetically originate from tryptamine and secologanin via an enzymatic Pictet-Spengler reaction and sequential diversification[19–27], while most of the chemical synthesis of these molecules uses the same Pictet-Spengler strategy, and these protocols dictate the functionality on C-1 position (Fig. 1b). Respecting the γ-tetrahydrocarbolines, which are less abundant in natural isolates, the most facile synthetic approach is the Fischer-indole reaction (Fig. 1b)[28]. Although the power of the

aforementioned methods is undisrupted, these protocols still have limitations, including normally tedious synthetic procedures, and limited diversity in piperidine fragments. And more to be considered is that in the course of structure-activity relationship studies in medicinal chemistry, these alkaloids-relevant molecular optimization has to rely on long-step modifications wherein some designed molecules are synthetically unreachable. Given the prevalence of indole alkaloids and the limitations of present synthetic methods, the development of modular synthesis of tetrahydrocarboline type of indole alkaloids with broad diversity in a rapid and efficient manner would greatly enrich the toolbox of organic chemists, enlarge the compound library available for drug discovery and also accelerate the lead compound optimization process.

The rapid assembly of complex scaffolds in a single step from simple precursors identifies as an ideal reaction in terms of efficiency and sustainability. Typically, multicomponent reactions (MCRs) fall into this ideal reaction and have become powerful synthetic tools for

[1]Institute of Drug Discovery and Design, National Key Laboratory of Advanced Drug Delivery and Release Systems, College of Pharmaceutical Sciences, Zhejiang University, Hangzhou 310058, China. ✉e-mail: slcui@zju.edu.cn

**Fig. 1 | Modular assembly of indole alkaloids through multicomponent reaction. a** Representative approved drugs with the structure of tetrahydrocarboline type of indole alkaloids. **b** Conventional synthetic approaches to tetrahydrocarboline type of indole alkaloids. **c** Modular assembly of tetrahydrocarboline type of indole alkaloids delineated in this work.

the construction of complex and diverse chemical entities in a one-pot fashion from three or more starting materials[29–32]. Compared to conventional stepwise synthesis, the MCRs could rapidly assemble complex molecules with simple manipulation, high atom- and step-economy; therefore, they have found widespread applications in the facile synthesis of natural products, market drugs, biologically active compounds, and materials. The prominent MCRs, such as Ugi, Mannich, Strecker, Biginelli reaction, or Hantzsch dihydropyridine synthesis, have been robust preparative drug discovery methods[33–44]. However, the MCRs-promoted modular and straightforward synthesis of tetrahydrocarboline type of indole alkaloids still remains explored. On the other hand, most of the reported MCRs mechanistically relied on those activated carbon centers to render the multiple chemical bond formations and target molecules construction, whereas the invoking of unactivated carbon centers poses a substantial challenge.

Bearing these challenges in mind and inspired by the Mannich reaction for the biomimetic synthesis of tropinone alkaloids, as well as our previous studies on MCRs[45–47], we speculated that if a functional group was incorporated into the *N*-atom of indole, the electron density distribution of indole moiety would be changed and this might have the opportunity to invoke the tethered unactivated carbon centers to involve in an MCR process toward the construction of indole alkaloids. Herein, we would like to report our successful implementation of this concept for the modular assembly of tetrahydrocarboline type of indole alkaloids through multicomponent reaction (Fig. 1c). This practical reaction could tolerate a broad range of functional groups so

that a wide set of market drugs and drug-like alkaloids could be achieved in a single-step from simple precursors, and this could also greatly enlarge the alkaloid library and much simplified the optimization process in drug discovery.

## Results

### Reaction optimization

We commenced our study by choosing the model substrates of indole derivatives **1**, formaldehyde **2**, and methyl glycine ester hydrochloride **3a**. Each reaction parameter was examined including solvent, temperature, and the *N*-substitution of indole (Table 1, for details, see Supplementary information), and we identified that an intriguing multicomponent reaction of *N*-benzyl-2-methyl indole **1a, 2**, and **3a** in the solvent of *N,N'*-dimethylformamide (DMF) under air could well occur to furnish γ-tetrahydrocarboline **4a** in excellent yield (92%). This meant it has overcome the intrinsic challenge of modular assembly of tetrahydrocarboline type of indole alkaloids in a single step, wherein multiple chemical bond formations were observed including an unactivated carbon center involved C(sp$^3$)–C(sp$^3$) connection. Besides, the *N*-substitution was important in this process wherein the alkyl groups were optimal while the hydrogen atom and electron-withdrawing groups (*t*-butoxycarbonyl, Boc; *p*-toluenesulfonyl, Ts) would lead to diminished reaction efficiency (Supplementary Table 2). This is probably because the *N*-substitution of indole would render drastic changes to the electron density distribution of indole moiety which is essential to this MCR process and the alkaloid formation.

## Table 1 | Reaction optimization[a]

| Entry | 2 (equiv.) | 3a (equiv.) | Solvent | Temp. | Time | Yield (%)[b] |
|---|---|---|---|---|---|---|
| 1 | 5 | 2 | DMSO (1 mL) | r. t. | 12 hr | 84 |
| 2 | 5 | 2 | DMF (1 mL) | r. t. | 12 hr | 92 |
| 3 | 5 | 2 | THF (1 mL) | r. t. | 12 hr | 87 |
| 4 | 5 | 2 | MeCN (1 mL) | r. t. | 1.5 hr | 25 |
| 5[c] | 5 | 2 | MeCN (1 mL) | 45 °C | 6 hr | trace |
| 6 | 5 | 2 | MeOH (1 mL) | r. t. | 12 hr | 12 |
| 7 | 5 | 2 | DMF (1 mL) | r. t. to 60 °C | 2 hr | 90 |
| 8 | 5 | 2 | DMF/MeCN (1 mL, 1:1) | r. t. to 45 °C | 2 hr | 90 |

[a]The reaction was conducted with 1a (0.15 mmol, 1 equiv.), formaldehyde 2 (37% in water, 0.75 mmol, 60 µL, 5 equiv.), 3a (0.3 mmol, 2 equiv.).
[b]Yield refers to isolated product by column chromatography on silica gel eluted with petroleum ether/ethyl acetate (v/v, 4:1).
[c]Polyformaldehyde was used as formaldehyde source.

### Reaction scope study and tetrahydrocarboline synthesis

We next focused on the generality of this multicomponent one-pot synthesis of indole alkaloids using this operationally simple protocol. Firstly, a wide set of amine-building blocks were explored (Fig. 2). For instance, diverse amino acid derivatives could well engage in this process with 2-methyl indoles 1 and formaldehyde 2 to deliver the corresponding products 4a–4l, and the chirality and functionality from amino acid could be directly incorporated into the indole alkaloids which could not be accessed by conventional methods. Besides, numerous flexible aliphatic amines were compatible with this protocol to furnish the indole alkaloids in good to excellent yields (4m–4ak), with a diversity of substituents attached, including chloride, bromide, sulfone, trifluoromethyl, cyano, pathalimide, acetoxy, alkene, alkyne, three- to six-membered cyclic and heterocyclic rings, phenol. Notably, these functionalities were widely used in drug discovery and they could be intuitively mapped onto the indole alkaloid frameworks from simple materials in this protocol. The structures (4f, 4af) were unambiguously confirmed by single-crystal x-ray crystallography. We next examined the compatibility of this protocol with indole building blocks. A variety of substitutions in the indole scaffold were well tolerated in this multicomponent synthesis of tetrahydrocarboline alkaloids, including fluoro, chloro, bromo, ester, methyl, phenyl, and pinacolboranyl (4al-4av). Notably, the 7-azaindole, a bioisostere of indole, was also applicable in this protocol (4aw). Meanwhile, various substitutions in the N-position of indole were varied, such as methyl, para-methoxybenzyl (PMB), 2-phenylethyl, allyl, cinnamyl, and tert-butyl carbonate (Boc) (4ax-4aac). Besides, we were pleased to find that the multi-substituted pyrroles were also applicable in this protocol for leading to pyrrole-piperidine fused products (4aad and 4aae). These functionalities pervade the structure-activity relationship studies in medicinal chemistry and serve as a productive springboard for further skeletal modification strategies.

Encouraged by the above results, we subsequently wondered whether β-tetrahydrocarboline could be assembled in the same manner using this MCR protocol (Fig. 3). Thus, the 3-methyl indole was employed as a building block for investigation, and the reaction parameters were examined including the substituents of indole, solvent, and reaction temperature. Gratifyingly, (Boc)2O, a widely used protecting group, was placed on the N-position of 3-methyl-5,6-dimethoxy indole (5) to render the occurrence of MCR with formaldehyde 2 and amine-building block 3. The generality was also examined, and a wide range of functionalities in the amines were well applicable in this process, including amino acid derivatives, alkene, alkyne, chloro, sulfone, cyano, phthalimide, and cyclobutane (6a–6n). Moreover, the prolonged aliphatic chain in the indole moiety with the functionality of ester and protected alcohol were also compatible to deliver the alkaloid products smoothly (6o–6p). Besides, a single group substitution in the indole skeleton such as methoxy, benzyloxy, or hydroxyl, would lead to diminished yields, probably due to the changed electron density distribution; however, when p-toluenesulfonic acid (p-TSA) was added as an additive, the corresponding β-tetrahydrocarbolines could be achieved in moderate to excellent yields (6q–6af; for details, see Supplementary information). In particular, We found that the reaction was dramatically affected by the N-substitutions, wherein the electron-withdrawing groups like benzyloxycarbonyl (Cbz) and N,N-dimethylformyl were optimal (6ag, 6ah, for details, see Supplementary information) and electron-donating groups were inferior. The structure (6v) was unambiguously confirmed by single-crystal x-ray crystallography. Considering the remarkable value of tetrahydrocarboline type of indole alkaloid in natural products and drug discovery, this protocol provides a rapid and distinct approach to new structured indole alkaloids available for drug discovery, and would definitely simplify the optimization process of lead molecule thus improving the accuracy and efficiency of candidate identification.

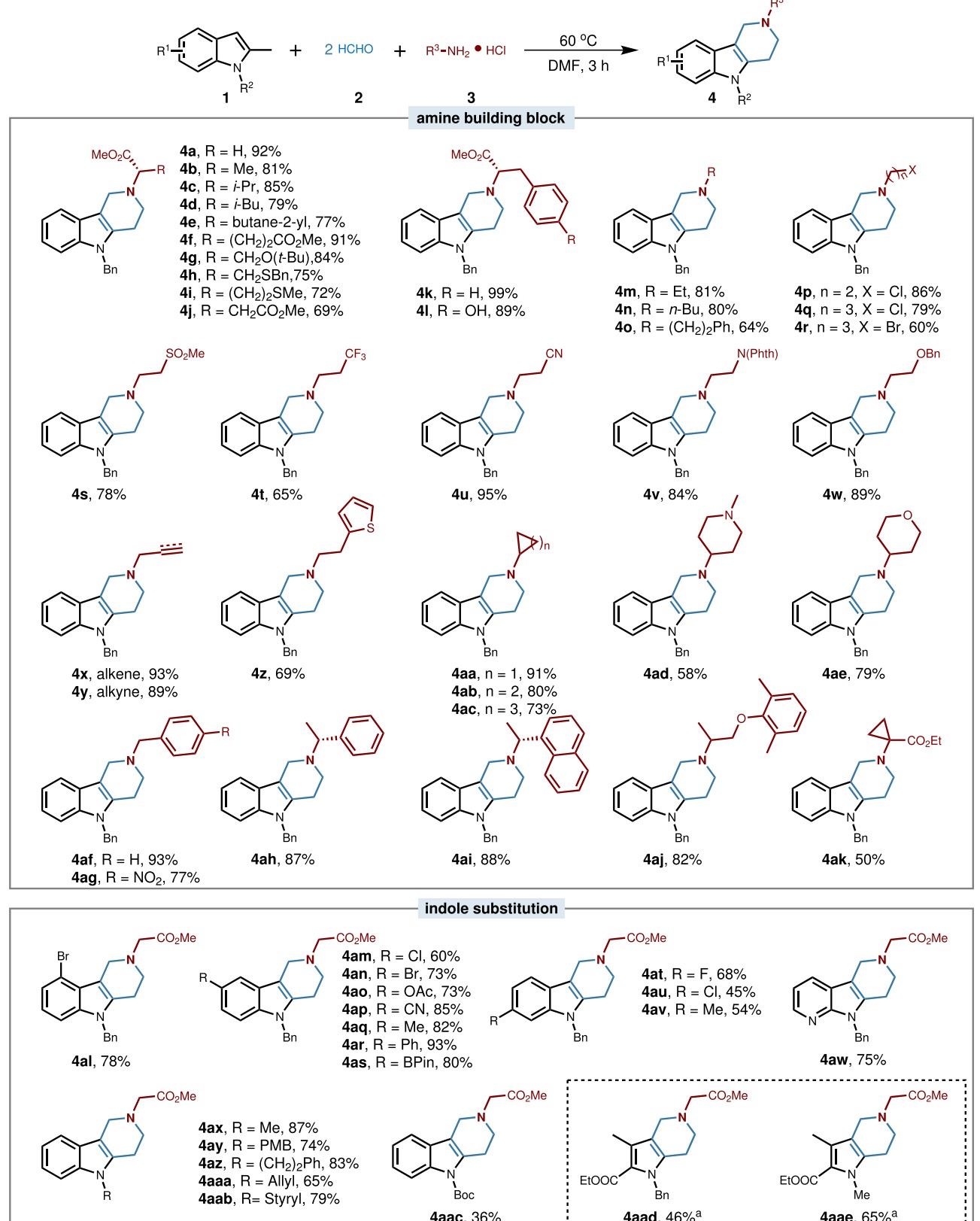

**Fig. 2 | Scope for the multicomponent synthesis of γ-tetrahydrocarboline type of indole alkaloids.** Conditions: **1** (0.2 mmol), **2** (1.0 mmol), **3** (0.4 mmol), DMF (1.5 mL), at 60 °C for 3 h. Isolated yields are given. Phth pathalimide, PMB *para*-methoxybenzyl, Boc *tert*-butyl carbonate. [a]CH₃CN (1.5 mL) was used as solvent, TsOH (0.1 mmol) was added as an additive and the reaction was conducted at 80 °C.

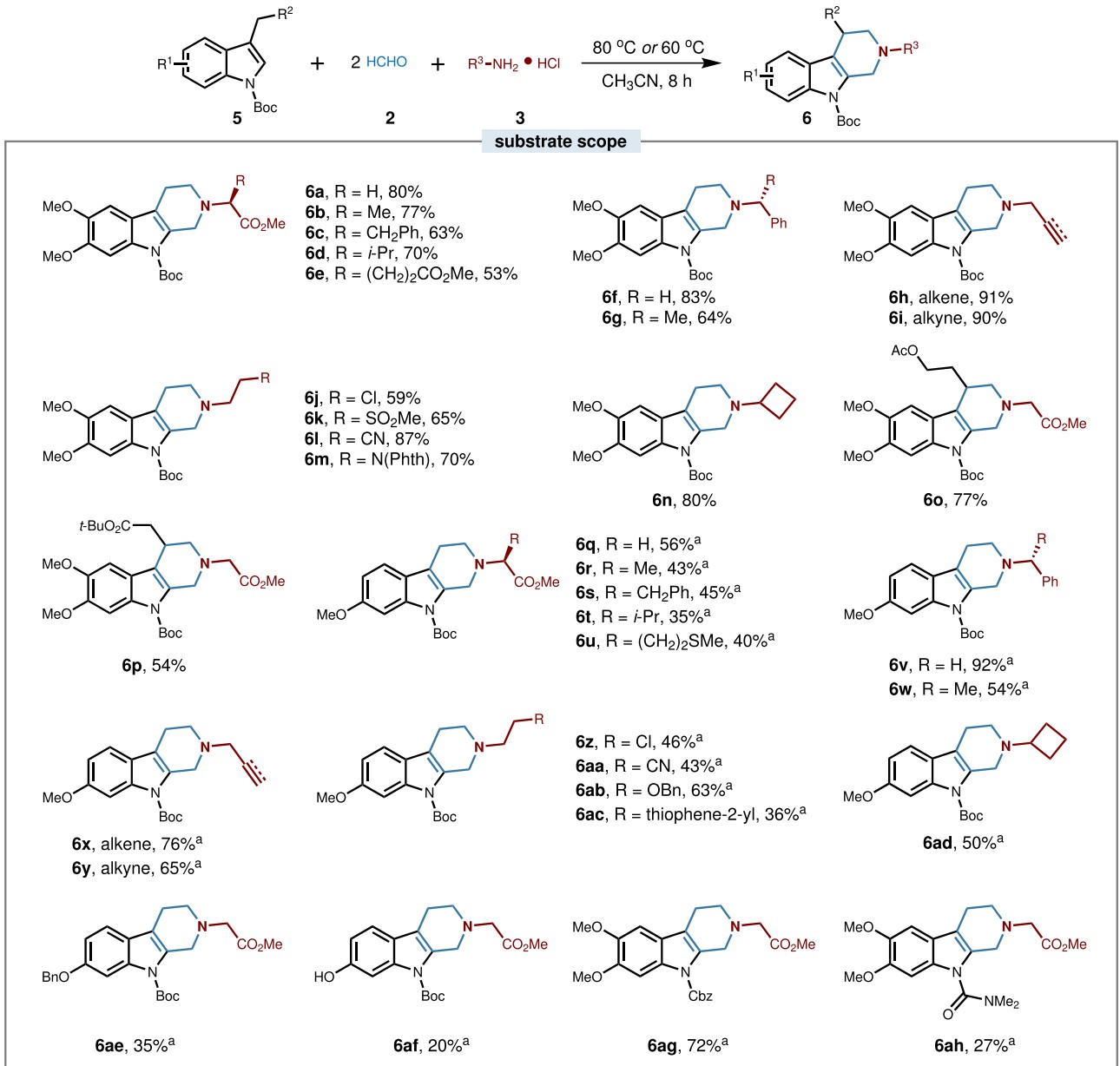

**Fig. 3 | Scope for the multicomponent synthesis of β-tetrahydrocarboline type of indole alkaloids.** Conditions: **1** (0.2 mmol), **2** (1.0 mmol), **3** (0.4 mmol), CH₃CN (1.5 mL), at 80 °C for 8 h. Isolated yields are given. Phth pathalimide, Boc *tert*-butyl carbonate. ᵃTsOH (0.1 mmol) was added as an additive and the reaction was conducted at 60 °C.

To highlight the value of this protocol, a collective and concise synthesis of market drugs and clinical candidates was exemplified (Fig. 4). For instance, Mebhydrolin (**8**) and γ-tetrahydrocarboline block (**9**), with the key structure of γ-tetrahydrocarboline, could be rapidly achieved from the feedstock 2-methyl indole (**7a**) in short step-synthetic routes under this protocol; while AVN-101 (**10**) and Dimebolin (**11**) could also be accessed from feedstock **7 g** in only two-step synthesis under this protocol; Setipiprant (**13**) could also be rapidly achieved under this strategy. A 5 mmol scale reaction was established to demonstrate the practicability. Meanwhile, various new structured β-tetrahydrocarboline types of drug-like compounds (**14–17**) could be rapidly assembled and further merged with peptide and click chemistry, which could occupy new chemical space and biological space for drug discovery. Moreover, once a structure optimization of indole alkaloids is needed, this protocol provides a distinctive shortcut rather than a long-step modification.

## Mechanistic investigations

Since the modular assembly of indole alkaloids and the unusual C(sp³)−C(sp³) connection occurrence in this process is impressive, the reaction mechanism was probed (Fig. 5). Control reaction, deuterium-labeling reaction, and cross-over reaction were set up to elucidate a multiple alkylamination cascade in this process[48–50]. For instance, the indole starting material **1a**, the mono-alkylaminated intermediate **18**, the di-alkylaminated intermediate **19**, were individually subject to the reaction with deuterated formaldehyde (DCDO, [D₂]-**2**) and glycine methyl ester hydrochloride **3a**, and the equivalent amount of [D₂]-**2** and **3a** was varied, and the results showed that the deuterated γ-tetrahydrocarboline **4a** ([D₄]-**4a**) could be formed with different deuteration ratio (Fig. 5a). Besides, when **18** was subjected to react with formaldehyde in the absence of **3a**, there was not any product formation (Fig. 5b). Interestingly, either **18** or **19** was subjected to the reaction with formaldehyde and benzylamine hydrochloride (**3af**),

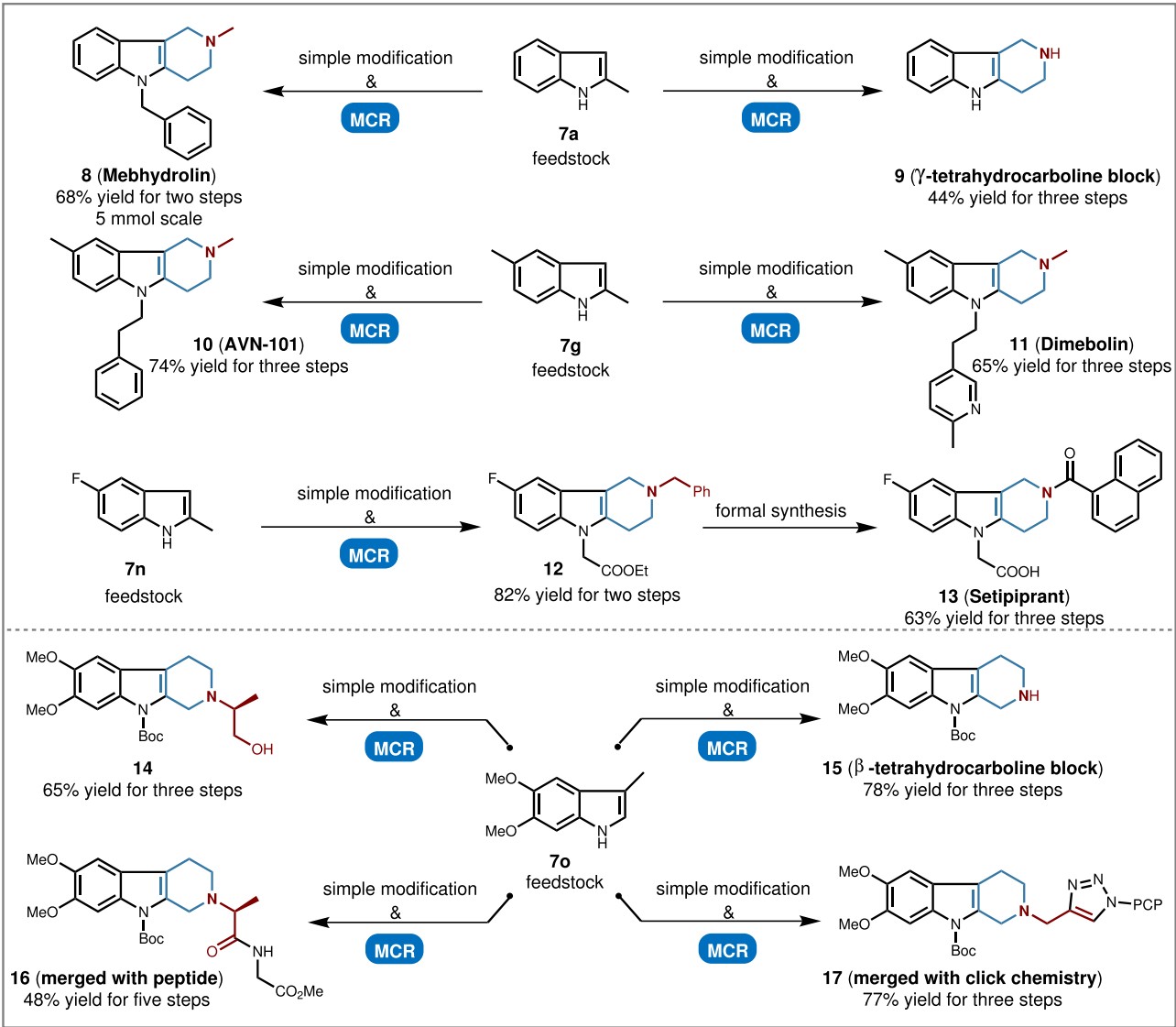

**Fig. 4 | Applications to the concise synthesis of market drugs and drug-like molecules.** MCR multicomponent reaction, PCP *para*-cyanophenyl.

**4af** was formed as the major product rather than **4a**. Meanwhile, a tertiary amine substituted indole **20** was prepared and subjected to reaction with **2** and **3a**, the reaction did not deliver the γ-tetrahydrocarboline product. Additionally, kinetic isotope effect (KIE) studies were carried out and a KIE value of 3.25 was obtained, indicating that the C(sp³)−H bond cleavage might be involved in the rate-limiting step (Fig. 5c, for details, see Supplementary information). Taking together these results, a plausible reaction mechanism was proposed (Fig. 5d). At beginning, the indole ring triggered the iterative alkylaminations to deliver **Int-2** and **Int-3**. Then, either **1a** or **Int-2** would undergo another alkylamination to deliver **Int-4**. At this stage, a further alkylamination via transition state **TS-1** would induce the unactivated carbon center to undergo dehydrogenation and convert the unactivated C(sp³) to activated C(sp²) along with the formation of **Int-5**. The following iminium formation of **Int-6**, cyclization, and retro-Mannich reaction would achieve the construction of alkaloids and enable the cascade cycle[51].

Similarly, control reaction and deuterium-labeling reaction were also conducted to investigate the mechanism for β-tetrahydrocarboline formation. For instance, the indole starting material **5a**, the mono-alkylaminated intermediate **22**, and the di-alkylaminated intermediate **23**, were individually subject to the

reaction with deuterated formaldehyde (DCDO, [D₂]-**2**) and glycine methyl ester hydrochloride **3a**, and the results showed that the deuterated γ-tetrahydrocarboline [D₄]-**6a** could be formed with almost completed deuteration ratio (Fig. 6a). Besides, the one-pot reaction of **22**, formaldehyde and benzylamine hydrochloride (**3af**) was set up, wherein **6f** was formed as the major product rather than **6a** (Fig. 6b). Additionally, kinetic isotope effect (KIE) studies were carried out and a KIE value of 2.64 was obtained, indicating that the C(sp³)−H bond cleavage might be involved in the rate-limiting step (Fig. 6c, for details, see Supplementary Information). These results indicated that the synthetic mechanism of β-tetrahydrocarboline is probably similar with γ-tetrahydrocarboline (Fig. 6a−c) and a plausible mechanism was then proposed (Fig. 6d). Initially, the indole ring triggered the iterative alkylaminations to deliver **Int-7** and **Int-8**. Then, either **5a** or **Int-7** would undergo another alkylamination to deliver **Int-9**. At this stage, a further alkylamination via transition state **TS-2** would induce the unactivated carbon center to undergo dehydrogenation and convert the unactivated C(sp³) to activated C(sp²) along with the formation of **Int-10**. The following iminium formation of **Int-11**, cyclization, and retro-Mannich reaction would achieve the construction of β-tetrahydrocarboline products **6** and enable the cascade cycle[51].

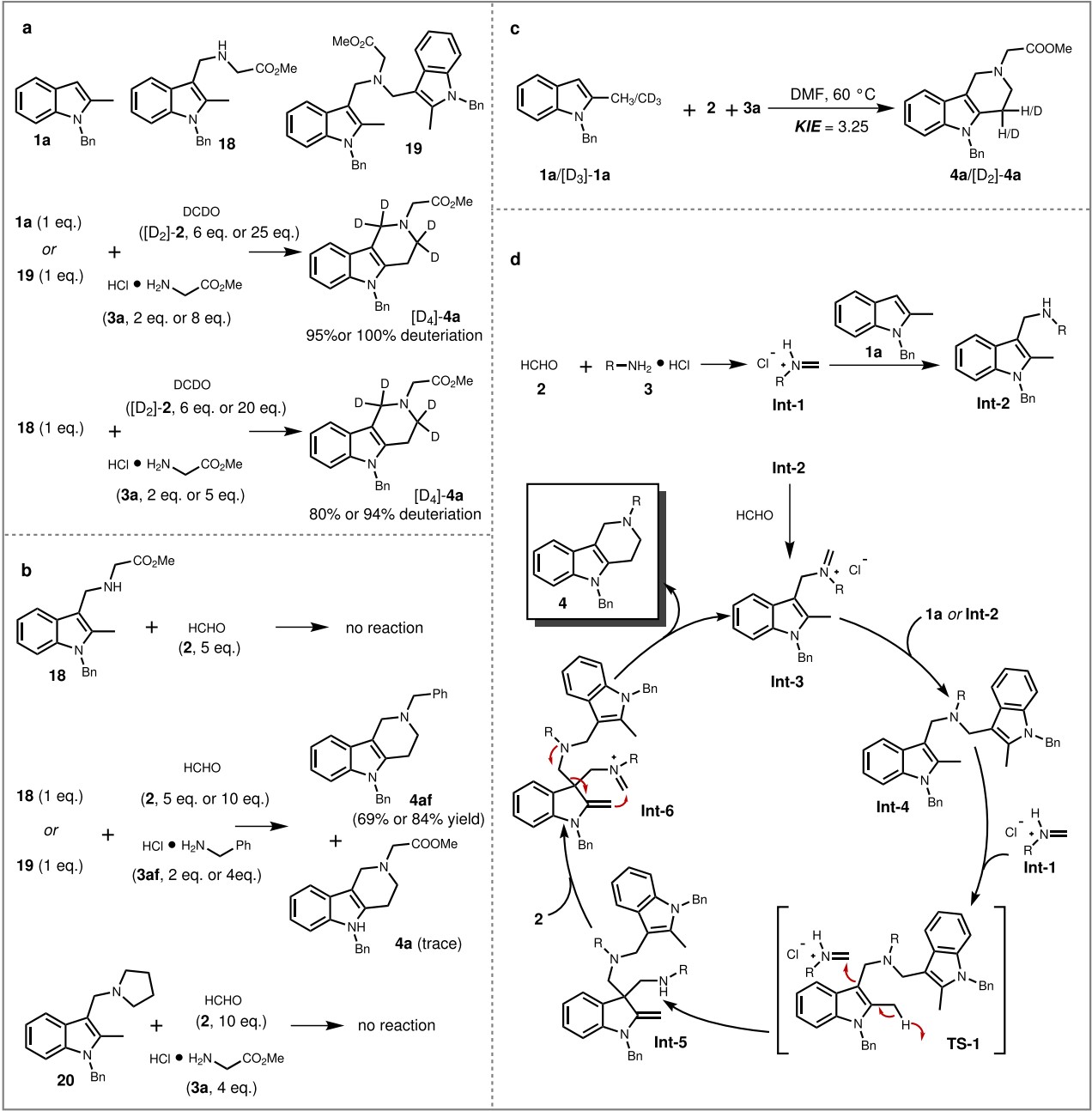

**Fig. 5 | Mechanistic studies for synthesis of γ-tetrahydrocarboline. a** Deuterium-labeling and stepwise control reaction. **b** Cross-over reaction. **c** Kinetic isotope effect experiments. **d** Proposed mechanism.

## Discussion

In conclusion, we have achieved a modular assembly of tetra-hydrocarbolines through multicomponent reaction of 2-substituted or 3-substituted indoles, formaldehyde, and amino hydrochlorides, and this protocol provides expedient access to these indole alkaloids. The chirality and a wide scope of functional groups were compatible in this process thus significantly enlarging the chemical space and biological space of tetrahydrocarbolines. Because tetrahydrocarbolines are widely encountered as pharmaceutically relevant substrates, we believe that this protocol would be a widely applicable strategy in indole-related drug discovery. Besides, this work also provides a new vision for the reaction of unactivated C(sp³) center and C(sp³)–C(sp³) bond formation.

## Methods

### General procedure for the synthesis of γ-tetrahydrocarboline 4a-4aac

A mixture of **1** (0.2 mmol), formaldehyde **2** (37% in water, 0.08 mL, 5 equiv.), and corresponding primary amine hydrochloride **3** (0.4 mmol, 2 equiv.) in DMF (1.5 mL) was stirred at 60 °C until the reaction was completed. The reaction was quenched by saturated aqueous NaHCO₃. The aqueous layer was extracted with ethyl acetate (three times), and the combined organic layer was dried over Na₂SO₄ and concentrated. Purification by silica gel column chromatography to give corresponding γ-tetrahydrocarboline products **4a**–**4aac**.

**Fig. 6 | Mechanistic studies for synthesis of β-tetrahydrocarboline. a** Deuterium-labeling and stepwise control reaction. **b** Cross-over reaction. **c** Kinetic isotope effect experiments. **d** Proposed mechanism.

## General procedure for the synthesis of 4aad and 4aae

A mixture of **1t** (or **1u**) (0.2 mmol), formaldehyde **2** (37% in water, 0.08 mL, 5 equiv.), TsOH (0.1 mmol, 0.5 equiv.) and methyl glycinate hydrochloride **3a** (0.4 mmol, 2 equiv.) in MeCN (1.5 mL) were stirred at 80 °C until the reaction was completed. The reaction was quenched by saturated aqueous NaHCO$_3$. The aqueous layer was extracted with ethyl acetate (three times), and the combined organic layer was dried over Na$_2$SO$_4$ and concentrated. Purification by silica gel column chromatography to give products **4aad** and **4aae**.

## General procedure for the synthesis of β-tetrahydrocarboline 6a–6p

A mixture of **5** (0.2 mmol), formaldehyde **2** (37% in water, 0.08 mL, 5 equiv.), and corresponding primary amine hydrochloride **3** (0.4 mmol, 2 equiv.) in MeCN (1.5 mL) was stirred at 80 °C until the reaction was completed. The reaction was quenched by saturated aqueous NaHCO$_3$. The aqueous layer was extracted with ethyl acetate (three times), and the combined organic layer was dried over Na$_2$SO$_4$ and concentrated. Purification by silica gel column chromatography to give corresponding β-tetrahydrocarboline products **6a–6p**.

## General procedure for the synthesis of 6q–6ah

A mixture of **5** (0.2 mmol), formaldehyde **2** (37% in water, 0.08 mL, 5 equiv.), TsOH (0.1 mmol, 0.5 equiv.) and corresponding primary amine hydrochloride **3** (0.4 mmol, 2 equiv.) in MeCN (1.5 mL) was stirred at 60 °C until the reaction was completed. The reaction was quenched by saturated aqueous NaHCO$_3$. The aqueous layer was extracted with ethyl acetate (three times), and the combined organic layer was dried over Na$_2$SO$_4$ and concentrated. Purification by silica gel column chromatography to give corresponding products **6q–6ah**.

## Data availability

The data that support the findings of this study are available in the main text or the supplementary materials. The X-ray crystallographic coordinates for structures reported in this study have been deposited at the Cambridge Crystallographic Data Centre (CCDC), under deposition numbers 2211850, 2211851 and 2235528. These data can be obtained free of charge from The Cambridge Crystallographic Data Centre via www.ccdc.cam.ac.uk/data_request/cif.

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

## Acknowledgements

We thank Zhenjun Mao (Department of Chemistry, Zhejiang University) and Jianyang Pan (Research and Service Center, College of Pharmaceutical Sciences, Zhejiang University) for performing NMR and HRMS spectrometry. We thank Jiyong Liu (Department of Chemistry, Zhejiang University) for performing the x-ray analysis. We are grateful for financial support from the NSFC (21971222, 22277106), the National Program for Support of Top-notch Young Professionals (grant 2022), the Natural Science Foundation of Zhejiang Province (Distinguished Young Scholar Program, LR23H300001), the Zhejiang Provincial Key R&D Program (grant 2023C03118).

## Author contributions

S.C. conceived and directed the project. Experiments and data analysis were conducted by J.L., W.Z., Z.L., and L.Z., S.C. wrote the manuscript.

## Competing interests

The authors declare no competing interests.
