## [Peer Review File · Nature Communications]

Modular Assembly of Indole Alkaloids Enabled by Multicomponent ReactionReviewers' Comments:

Reviewer #1:

Remarks to the Author:

Indole alkaloids especially γ -tetrahydrocarbolines are important structural motifs possessed in numerous biological activity molecules, pharmaceuticals and agricultural chemicals. Despite its significance, synthesizing tetrahydrocarboline still limited in tedious synthetic procedures and piperidine fragments diversity. In this manuscript, Cui and coworkers report a modular assembly of tetrahydrocarboline type of indole alkaloids using 2-substituted or 3-substituted indoles, formaldehyde and amino derivative hydrochlorides in a single step, delivering a series of γ - tetrahydrocarbolines or β -tetrahydrocarbolines in high yields under mild reaction conditions. The site selectivity of the reaction was switched by simply changing the position of indole substituents leading to different tetrahydrocarbolines, which was quiet exciting. Furthermore, this method is applied to rapid access to pharmaceutically interesting compounds such as AVN-101, dimebolin and setipiprant. Control reaction, deuterium-labeling reaction, and cross-over reaction elucidate the reaction mechanism as a multiple alkylamination cascade reaction. Overall, the manuscript is well organized and ESI has high quality. I support its publication in Nature Communication after major revision.

1. Reaction mechanism involving 3-substituted indole for the synthesis of β -tetrahydrocarbolines needed to be placed in the manuscript. And discussion of the N-substitution should also be included.
 2. In control experiments in Fig. 6b, no reaction for 18 with formaldehyde, and no reaction of 20 with formaldehyde and amine. According to the proposed mechanism, transformation into analogues of int 3 (from 18 and aldehyde) or int-5 and product 4 (from int-4 and imine salt) should be detected because the NRR' group leave after formation of tetrahydrocarboline. The possible transformations should be checked carefully. Otherwise, the role of 1-(2-methyl-1H-indol-3-yl)methanamine unit should be discussed in detail.
 4. As for 3-substituted indole, the scope of the transformation is broad but limited to methoxy substituted and N-Boc protected indole. It would be interesting to see if authors try substrates with different substituents.
 5. What is the rate limiting step of this reaction? The reaction kinetics need to be investigated. Formation of byproduct 21 should be discussed.
 6. Although indole alkaloids are important structural motifs, other heterocyclic compound such as pyrrole, furan, benzofuran, thiophen, benzo thiophen, and others should be investigated.
 7. Whether mixed formaldehyde and other aldehyde could deliver substituted tetrahydrocarboline?
- Other suggestions:

1. The Chemdraw need to be polished to gain good esthetical effect. For example, the color scheme being used throughout the manuscript could be improved. The chemdraw on the mechanism part could be added more color to distinguish each part and well organized. With respect to the substrate scope part, the placement of products could be more aesthetically arranged.
2. "The prominent MCRs, such as Ugi, Mannich, Strecker, Biginelli reaction, or Hantzsch dihydropyridine synthesis, have been robust preparative drug discovery methods." References need to be included in this part.
3. Please carefully check through the main text, references and ESI, there are a number of typos errors and inconsistence.

Reviewer #2:

Remarks to the Author:

The paper of S. Cui and co-workers reports an interesting modular assembly of indole alkaloids enabled by multicomponent reactions. As indole core is among the most important and abundant scaffolds in natural products, in particular alkaloids, these MCRs leading either to β -, and γ -tetrahydrocarbolines represents valuable pathways towards the indole-based skeletons. Interestingly, this work provides a new vision for the reaction of unactivated C(sp³) center and C(sp³)-C(sp³) bond

formation. Even though the manuscript is well written, the scope of the MCRs well explored, and the applicability of these procedures in the synthesis of market drugs widely demonstrated, there are some open questions and points that need to be clarified (see below). I recommend rejection and resubmission after the following points have been addressed in the revised version:

1. While the mechanism relating to the formation of γ -carboline has been addressed, the authors did not indicate on the one related to the formation of β -carboline. Given the intrinsic reactivity of indole substrate, the mechanism of the multicomponent reaction that leads to the formation of β -carboline cannot be mirrored to that of formation of γ -carboline. Therefore, this work requires a more detailed mechanism study into the β -carboline formation before it would benefit the reader of Nature Communications.

2. The mechanistic scheme of the reaction is crucially dependent on the formation of Int-4 (19) According to the cross-over experiment described in Fig. 5b, the reaction of 18, formaldehyde and benzylamine hydrochloride (3af) delivered the γ -tetrahydrocarboline 4af rather than 4a. How do the authors explain the formation of the first intermediate reported in bracket (6.4 in Supp. Inf.) since the starting indole 1a is not present in this experiment? On the other hand, if the path a of Fig. S3 is operative, a mixture of products 3af and 4af should be obtained. Could be possible that the authors used the substrate 19 instead of 18 for this experiment? This point should be better clarified.

Minor modifications like these:

Page 5: "1d" should be "1a".

Page 5: the authors state that N-unsubstituted indole led to diminished reaction efficiency. However, this example does not appear in Table S2 of Supp. Inf.

Page 6: in the footnote a of Table 1, please insert the amounts of formaldehyde and 3a used in these reactions.

Page 7: "4a-4ak" should be "4m-4ak".

Page 7: "ester" should be "acetoxy". The substituent "cyano" should be also included.

Page 7: "bioisosterism" should be "bioisostere".

Page 8: in the representative formula of compounds 4p-4r of Fig. 2, "Cl" should be replaced with "X".

Pages 9 and 10: in the caption of Fig. 2 and 3, "...2 (0.5 mmol),..." should be "...2 (1.0 mmol)...".

Page 11: "7b" should be "7g".

Page 12: "PCP = para-cynaophenyl" should be "PCP = para-cyanophenyl".

Page 14: "Discussion" should be "Conclusion".

Concerning the use of indoles in MCRs, some reviews should be cited in the references section.

In the Scheme of Page 62 of Supp. Inf. "7m" should be "7n".

Reviewer 1

(1) Reaction mechanism involving 3-substituted indole for the synthesis of β -tetrahydrocarbolines needed to be placed in the manuscript. And discussion of the *N*-substitution should also be included.

Response: Thank you for these suggestions. We have carried out many experiments to investigate the synthetic mechanism. The results were added in the manuscript and Fig 6. Besides, more details were added in Supplementary Information.

The comments added in the manuscript is listed as “Similarly, control reaction and deuterium-labelling reaction were also conducted to investigate the mechanism for β -tetrahydrocarboline formation. For instance, the indole starting material **5a**, the mono-alkylaminated intermediate **22**, the di-alkylaminated intermediate **23**, were individually subject to the reaction with deuterated formaldehyde (DCDO, [D₂]-**2**) and glycine methyl ester hydrochloride **3a**, and the results showed that the deuterated γ -tetrahydrocarboline [D₄]-**6a** could be formed with almost completed deuteration ratio (Fig. 6a). Besides, the one-pot reaction of **22**, formaldehyde and benzylamine hydrochloride (**3af**) was set up, wherein **6f** was formed as the major product rather than **6a** (Fig. 6b). Additionally, kinetic isotope effect (KIE) studies were carried out and a KIE value of 2.64 was obtained, indicating that the C(*sp*³)-H bond cleavage might be involved in the rate-limiting step (Fig. 6c, for details, see supplementary information). These results indicated that the synthetic mechanism of β -tetrahydrocarboline is probably similar with γ -tetrahydrocarboline (Fig. 6a-6c) and a plausible mechanism was then proposed (Fig. 6d). Initially, the indole ring triggered the iterative alkylaminations to deliver **Int-7** and **Int-8**. Then, either **5a** or **Int-7** would undergo another alkylamination to deliver **Int-9**. At this stage, a further alkylamination via transition state **TS-2** would induce the unactivated carbon center to undergo dehydrogenation and convert the unactivated C(*sp*³) to activated C(*sp*²) along with the”

formation of **Int-10**. The following iminium formation of **Int-11**, cyclization, and retro-Mannich reaction would achieve the construction of β -tetrahydrocarboline products **6** and enable the cascade cycle⁵¹.”

Fig. 6. Mechanistic studies for β -tetrahydrocarboline. a Deuterium-labeling and stepwise control reaction. **b** Cross-over reaction. **c** Kinetic isotope effect experiments. **d** Proposed mechanism.

Respecting to the *N*-substituents, we have screened these for the multicomponent synthesis of β -tetrahydrocarboline. And the results were presented in the manuscript and supplementary information. We found that the reaction was significantly affected by the *N*-substitutions. The electron-donating groups, such as methyl and benzyl, would lead to instability of the indole framework and fail to afford the products. In contrast, the electron-withdrawing groups like benzyloxycarbonyl (Cbz) and *N,N*-dimethylformyl substituents were well tolerated in this process to afford the corresponding β -tetrahydrocarboline products (**6ag**, **6ah**).

The screen of *N*-substituents for the multicomponent synthesis of β -tetrahydrocarbolines are added as Table S6 (page 29) in supplementary information.

Table S6. Screen the substituents for the multicomponent synthesis of β -tetrahydrocarboline*

Entry	R ¹	R ²	R ³	Solvent	Temp.	Time	Yield (%)
1	OMe	OMe	Bn	MeCN (1 mL)	r. t. to 60 °C	2 hr	N/A
2	OMe	OMe	Me	MeCN (1 mL)	r. t. to 60 °C	2 hr	N/A
3	OMe	OMe	Boc	MeCN (1 mL)	r. t. to 80 °C	6 hr	80
4	OMe	OMe	Cbz	MeCN (1 mL)	r. t. to 80 °C	6 hr	65
5	OMe	OMe	H	MeCN (1 mL)	r. t. to 60 °C	2 hr	N/A
6	OMe	H	Boc	MeCN (1 mL)	r. t. to 80 °C	8 hr	40
7	OMe	H	Bn	DMF (1 mL)	r. t. to 60 °C	2 hr	trace
8^a	OMe	H	Boc	MeCN (1 mL)	r. t. to 60 °C	8 hr	56
9	H	OMe	Bn	DMF (1 mL)	r. t. to 60 °C	2 hr	N/A
10	H	H	Me	MeCN (1 mL)	r. t. to 60 °C	8 hr	N/A

*The reactions conducted with **4** (0.15 mmol, 1 equiv.), formaldehyde **2** (37% in water, 0.75 mmol, 60 μ L, 5 equiv.), **3a** (0.3 mmol, 2 equiv.) and yield refers to isolated product by column chromatography on silica gel eluted with petroleum ether/ethyl acetate (v/v, 2:1); ^aTsOH (0.075 mmol, 0.5 equiv.) was added as an additive.

(2) In control experiments in Fig. 6b, no reaction for **18** with formaldehyde, and no reaction of **20** with formaldehyde and amine. According to the proposed mechanism, transformation into analogues of **int 3** (from **18** and aldehyde) or **int-5** and product **4** (from **int-4** and imine salt) should be detected because the NRR' group leave after formation of tetrahydrocarboline. The possible transformations should be checked carefully. Otherwise, the role of 1-(2-methyl-1H-indol-3-yl)methanamine unit should be discussed in detail.

Response: Thanks for your nice suggestions, and these suggestions are very helpful. We have conducted many experiments to well elucidate the mechanism, and now it is very clear. These details are added in Supplementary Information (page 82-87).

Firstly, we subjected intermediate **18** or **19** to the reaction condition with formaldehyde individually, and did not observe the formation of **4a**. Interestingly, when we added 1 drop aqueous HCl solution (1 N) as an additive to the reaction solution of **18** and formaldehyde, TLC analyses showed that **19** and **4a** were formed.

At this stage, we hypothesized that an alkylation would occur from two molecules of **18** and formaldehyde to generate the intermediate **19** and release an imino group. Subsequently, cyclization, and retro-Mannich reaction would achieve the construction of **4a**.

In addition, the tertiary amine substituted indole **20** was subjected to the cross-over experiment. As expected, **4a** was not observed in the reaction, because the tertiary amine **20** could not form the di-alkylaminated intermediate (**Int-4**) thus failing to drive the following cascade.

(4) As for 3-substituted indole, the scope of the transformation is broad but limited to methoxy substituted and *N*-Boc protected indole. It would be interesting to see if authors try substrates with different substituents.

Response: Thanks for this good suggestion. For the indole ring substituents, the benzyloxy and hydroxyl substituted indoles were tested and the corresponding β -tetrahydrocarbolines **6ae** and **6af** were added in the manuscript. Respecting to the *N*-protecting group, many groups were tested and the electron-rich group were found inferior while the electron-withdrawing groups were found optimal. In particular, the *N*-benzyloxycarbonyl (Cbz) and *N*-(*N,N*-dimethyl)formyl group were found applicable, and the corresponding products (**6ag** and **6ah**) were added in the text.

The screen of *N*-substituents for the multicomponent synthesis of β -tetrahydrocarbolines are added as Table S6 (page 29) in Supplementary Information.

(5) What is the rate limiting step of this reaction? The reaction kinetics need to be investigated. Formation of byproduct **21** should be discussed.

Response: Thank you for nice suggestion and it is very helpful. We have conducted the kinetic isotopic effect experiments and added in the text, and more details were added in the Supplementary Information (page 94-96).

Entry 1: Due to the *KIE* value in **1a**/[D]-**1a** being much less than 1.0, we conducted stability experiments on the compound [D]-**1a**. Due to the high boiling point of DMF, the results have no obvious comparison. Therefore, the solvent was replaced with DCM. As shown in the Supporting Information Fig. S5, the compound [D]-**1a** could transform into **1a** spontaneously. Thus, the inaccurate *KIE* value of [D]-**1a** is due to its instability. The *KIE* value of [D₃]-**1a** is observed as 3.25, thus it can be seen that the C(*sp*³)-H bond cleavage might be involved in the rate-limiting step.

Entry 2: kinetic isotope effect (KIE) studies were carried out and a KIE value for [D]-**5a** and [D₃]-**5a** 2.64 were obtained as 1.27 and 2.64, indicating that the C(*sp*³)-H bond cleavage might be involved in the rate-limiting step (Fig. 6c, for details, see supplementary information).

Respecting the byproduct **21** and **24**, we have added this in the Supplementary Information. **21** and **24** are type of bis(indolyl)alkane compounds, and their formation rely on a mechanism of acid-catalyzed coupling of indoles with formaldehyde (ref. 11 and ref. 12 in Supplementary Information).

(6) Although indole alkaloids are important structural motifs, other heterocyclic compound such as pyrrole, furan, benzofuran, thiophen, benzo thiophen, and others should be investigated.

Response: Thanks for the suggestion. After various trial, we were pleased to find that the multi-substituted pyrroles were applicable for leading to the corresponding pyrrole-piperidine fused products (**4aad** and **4aae**). These results were added in the manuscript and **Fig 2**.

(7) Whether mixed formaldehyde and other aldehyde could deliver substituted tetrahydrocarboline?

Response: Thanks for the suggestion. We have evaluated the reactions with mixed formaldehyde and benzaldehyde. The results showed that benzaldehyde could not involve in this multicomponent reaction to deliver substituted tetrahydrocarboline.

Other suggestions:

(1) The Chemdraw need to be polished to gain good esthetical effect. For example, the color scheme being used throughout the manuscript could be improved. The chemdraw on the mechanism part could be added more color to distinguish each part and well organized. With respect to the substrate scope part, the placement of products could be more aesthetically arranged.

Response: Thank you for your nice suggestion. We have polished the Chemdraw and added color to distinguish each part.

(2) “The prominent MCRs, such as Ugi, Mannich, Strecker, Biginelli reaction, or Hantzsch dihydropyridine synthesis, have been robust preparative drug discovery

methods.” References need to be included in this part.

Response: Thank you for your suggestion. References 33-44 were added in this part.

(3) Please carefully check through the main text, references and ESI, there are a number of typos errors and inconsistency.

Response: Thank you for the suggestion, and we have checked the text, reference and supporting information carefully and corrected the errors.

Reviewer 2

(1) While the mechanism relating to the formation of γ -carboline has been addressed, the authors did not indicate on the one related to the formation of β -carboline. Given the intrinsic reactivity of indole substrate, the mechanism of the multicomponent reaction that leads to the formation of β -carboline cannot be mirrored to that of formation of γ -carboline. Therefore, this work requires a more detailed mechanism study into the β -carboline formation before it would benefit the reader of Nature Communications.

Response: Thanks for nice suggestion. We have conducted many experiments to investigate the synthetic mechanism of β -tetrahydrocarbolines. And the result was added in the manuscript and Fig. 6, while more details was represented in Supplementary Information (page 90-96). The comments added in the text was listed as following:

“Similarly, control reaction and deuterium-labelling reaction were also conducted to investigate the mechanism for β -tetrahydrocarboline formation. For instance, the indole starting material **5a**, the mono-alkylaminated intermediate **22**, the di-alkylaminated intermediate **23**, were individually subject to the reaction with deuterated formaldehyde (DCDO, $[D_2]$ -**2**) and glycine methyl ester hydrochloride **3a**, and the results showed that the deuterated γ -tetrahydrocarboline $[D_4]$ -**6a** could be formed with

almost completed deuteration ratio (Fig. 6a). Besides, the one-pot reaction of **22**, formaldehyde and benzylamine hydrochloride (**3af**) was set up, wherein **6f** was formed as the major product rather than **6a** (Fig. 6b). Additionally, kinetic isotope effect (KIE) studies were carried out and a KIE value of 2.64 was obtained, indicating that the C(sp³)-H bond cleavage might be involved in the rate-limiting step (Fig. 6c, for details, see supplementary information). These results indicated that the synthetic mechanism of β-tetrahydrocarboline is probably similar with γ-tetrahydrocarboline (Fig. 6a-6c) and a plausible mechanism was then proposed (Fig. 6d). Initially, the indole ring triggered the iterative alkylaminations to deliver **Int-7** and **Int-8**. Then, either **5a** or **Int-7** would undergo another alkylamination to deliver **Int-9**. At this stage, a further alkylamination via transition state **TS-2** would induce the unactivated carbon center to undergo dehydrogenation and convert the unactivated C(sp³) to activated C(sp²) along with the formation of **Int-10**. The following iminium formation of **Int-11**, cyclization, and retro-Mannich reaction would achieve the construction of β-tetrahydrocarboline products **6** and enable the cascade cycle⁵¹.”

Fig. 6. Mechanistic studies for β -tetrahydrocarboline. a Deuterium-labeling and stepwise control reaction. **b** Cross-over reaction. **c** Kinetic isotope effect experiments. **d** Proposed mechanism.

(2) The mechanistic scheme of the reaction is crucially dependent on the formation of **Int-4 (19)**. According to the cross-over experiment described in Fig. 5b, the reaction of **18**, formaldehyde and benzylamine hydrochloride (**3af**) delivered the γ -tetrahydrocarboline **4af** rather than **4a**. How do the authors explain the formation of the first intermediate reported in bracket (6.4 in Supp. Inf.) since the starting indole **1a** is not present in this experiment? On the other hand, if the path a of Fig. S3 is operative, a mixture of products **3af** and **4af** should be obtained. Could be possible that the authors used the substrate **19** instead of **18** for this experiment?

Response: Thanks for your nice suggestions, and these suggestions are very helpful. We have conducted many experiments to well elucidate the mechanism, and now it is very clear. These details are added in Supplementary Information (page 82-87).

Firstly, we subjected intermediate **18** or **19** to the reaction condition with formaldehyde individually, and did not observe the formation of **4a**. Interestingly, when we added 1 drop aqueous HCl solution (1 N) as an additive to the reaction solution of **18** and formaldehyde, TLC analyses showed that **19** and **4a** were formed.

At this stage, we hypothesized that an alkylation would occur from two molecules of **18** and formaldehyde to generate the intermediate **19** and release an imino group. Subsequently, cyclization, and retro-Mannich reaction would achieve the construction of **4a**.

In addition, the tertiary amine substituted indole **20** was subjected to the cross-over experiment. As expected, **4a** was not observed in the reaction, because the tertiary amine **20** could not form the di-alkylaminated intermediate (**Int-4**) thus failing to drive the following cascade.

On the other hand, **18** and **19** were also subjected to the cross-over reaction. The one-pot reaction of **18** (or **19**), formaldehyde and benzylamine hydrochloride would deliver the *N*-benzyl γ -tetrahydrocarboline **4af** as major product, while **4a** was formed in trace amount. Therefore, the reviewer's suggestion is much helpful.

A reasonable explanation was also listed below:

Minor modifications like these:

Page 5: “**1d**” should be “**1a**”.

Page 5: the authors state that N-unsubstituted indole led to diminished reaction efficiency. However, this example does not appear in Table S2 of Supp. Inf.

Page 6: in the footnote a of Table 1, please insert the amounts of formaldehyde and **3a** used in these reactions.

Page 7: “**4a-4ak**” should be “**4m-4ak**”.

Page 7: “ester” should be “acetoxy”. The substituent “cyano” should be also included.

Page 7: “bioisosterism” should be “bioisostere”.

Page 8: in the representative formula of compounds **4p-4r** of Fig. 2, “Cl” should be replaced with “X”.

Pages 9 and 10: in the caption of Fig. 2 and 3, “...**2** (0.5 mmol),...” should be “...**2** (1.0 mmol)...”.

Page 11: “**7b**” should be “**7g**”.

Page 12: “PCP = para-cynaophenyl” should be “PCP = para-cyanophenyl”.

Page 14: “Discussion” should be “Conclusion”.

Concerning the use of indoles in MCRs, some reviews should be cited in the references section.

In the Scheme of Page 62 of Supp. Inf. “**7m**” should be “**7n**”.

Response: Thank you for the suggestions, all these errors have been corrected and we have also checked the text carefully. The reviews for indole related MCRs have been added as ref. 31 and ref. 32.

Reviewers' Comments:

Reviewer #1:

Remarks to the Author:

The authors have carefully studied the reaction mechanism as suggested, and now it is clearer. The MCR reaction could be extended to pyrrole. It is recommended to be accepted after tiny check. The structure of the product 6a in Figure 6a is incorrect (N is missing)

Reviewer #2:

Remarks to the Author:

With respect to the previous submission (NCOMMS-23-14169) the authors have addressed all the issues indicated by the previous referees including the discussion of the reaction mechanism relating to the formation of β -carboline. Also the number of compounds has been increased, expanding the scope of the study. Now, the quality of the manuscript is enhanced and therefore the reported research is suitable for being published in Nature Communications.

There are still some minor typographic type errors, but these are small revisions only for the editorial process.

Additional questions/suggestions:

- Further informations could be useful in order to elucidate the mechanisms of these multicomponent transformations. According to the cross-over experiment described in Fig. 5b (6.4 on Supp. Inf.), the reaction of 18 (or 19), formaldehyde and benzylamine hydrochloride 3af delivered the γ -tetrahydrocarboline 4af rather than 4a. On the other hand, compound 4a formation would be expected by conducting the reaction between the corresponding benzyl amino derivative of 18 (or 19), methyl glycine ester hydrochloride 3a and formaldehyde 2.

The same argument should also valid for the formation of β -tetrahydrocarboline (Fig. 6b, 6.7 on Supp. Inf.) starting from the benzyl amino analogue of compound 22 (or 23). I suggest the authors to verify it.

- In the Table S2 and S6 the term "N/A" is used. What does it really mean? N/A = not available? Please specify it.

- Since the Mannich N-indolylmethylation of amino acid derivatives (see the preparation of compounds 18, 19 and 20) is a reaction known in the literature, the relative reference should be added (Synthesis 2017, 49, 2257–2265).

- For compound 6 of the Scheme of Table S6 (p29_Supp. Inf. File), "R" should be "R3".

Reviewer 1

(1) The structure of the product **6a** in Figure 6a is incorrect (N is missing).

Response: Thank you for nice suggestion, this error has been corrected and we have also checked the text, reference and supporting information carefully and corrected the errors.

Reviewer 2

(1) There are still some minor typographic type errors, but these are small revisions only for the editorial process.

Response: Thank you for these suggestions, we have checked the text carefully and the errors have been corrected.

(2) Further informations could be useful in order to elucidate the mechanisms of these multicomponent transformations. According to the cross-over experiment described in Fig. 5b (6.4 on Supp. Inf.), the reaction of **18** (or **19**), formaldehyde and benzylamine hydrochloride **3af** delivered the γ -tetrahydrocarboline **4af** rather than **4a**. On the other hand, compound **4a** formation would be expected by conducting the reaction between the corresponding benzyl amino derivative of **18** (or **19**), methyl glycine ester hydrochloride **3a** and formaldehyde **2**.

The same argument should also valid for the formation of β -tetrahydrocarboline (Fig. 6b, 6.7 on Supp. Inf.) starting from the benzyl amino analogue of compound **22** (or **23**). I suggest the authors to verify it.

Response: Thanks for your nice suggestions, and these suggestions are very helpful. We have conducted these experiments according to this suggestion, and now the reaction mechanism is very clear. These details are added in Supplementary Information (page 89-95).

Firstly, **19-f** was subjected to the cross-over reaction. The one-pot reaction of **19-f**, formaldehyde and glycine methyl ester hydrochloride **3a** would deliver the γ -tetrahydrocarboline **4a** as major product, while **4af** was formed in trace amount.

On the other hand, **22-f** were also subjected to the cross-over reaction. As expected, the one-pot reaction of **22-f**, formaldehyde and glycine methyl ester hydrochloride **3a** would deliver the β -tetrahydrocarboline **6a** as major product.

Minor modifications like these:

In the Table S2 and S6 the term “N/A” is used. What does it really mean? N/A = not available? Please specify it.

Since the Mannich N-indolylmethylation of amino acid derivatives (see the preparation of compounds **18**, **19** and **20**) is a reaction known in the literature, the relative reference should be added (Synthesis 2017, 49, 2257 – 2265).

For compound **6** of the Scheme of Table S6 (p29_Supp. Inf. File), “R” should be “R³” .

Response: Thank you for the suggestions, and all these errors have been corrected and we have also checked the text carefully. The relative reference (Synthesis 2017, 49, 2257 – 2265) for preparation of compounds **18**, **19** and **20** has been added as ref. 13 in Supplementary Information.